# Differences in Perceived Risk of Contracting SARS-CoV-2 during and after the Lockdown in Sub-Saharan African Countries

**DOI:** 10.3390/ijerph182111091

**Published:** 2021-10-21

**Authors:** Uchechukwu Levi Osuagwu, Chikasirimobi G Timothy, Raymond Langsi, Emmanuel K Abu, Piwuna Christopher Goson, Khathutshelo P Mashige, Bernadine Ekpenyong, Godwin O Ovenseri-Ogbomo, Chundung Asabe Miner, Richard Oloruntoba, Tanko Ishaya, Deborah Donald Charwe, Esther Awazzi Envuladu, Obinna Nwaeze, Kingsley Emwinyore Agho

**Affiliations:** 1Translational Health Research Unit (THRI), School of Medicine, Western Sydney University, Campbelltown, NSW 2560, Australia; K.Agho@westernsydney.edu.au; 2Westville Campus, African Vision Research Institute, Discipline of Optometry, University of KwaZulu-Natal, Durban 3629, South Africa; mashigek@ukzn.ac.za (K.P.M.); bekpenyong@unical.edu.ng (B.E.); 3Department of Optometry and Vision Sciences, School of Public Health, Biomedical Sciences and Technology, Masinde Muliro University of Science and Technology, Kakamega 190-50100, Kenya; chikasirimobi@gmail.com; 4Health Division, University of Bamenda, Bambili P.O. Box 39, Bamenda 00237, Cameroon; raylangsi@yahoo.com; 5Department of Optometry and Vision Science, School of Allied Health Sciences, College of Health and Allied Sciences, University of Cape Coast, Cape Coast 00233, Ghana; eabu@ucc.edu.gh; 6Department of Psychiatry, College of Health Sciences, University of Jos, P.M.B. 2084, Jos 930003, Nigeria; piwunag@unijos.edu.ng; 7Department of Public Health, Faculty of Allied Medical Sciences, College of Medical Sciences, University of Calabar, Calabar 540271, Cross River State, Nigeria; 8Department of Optometry, Centre for Health Sciences, University of the Highlands and Islands, Inverness IV2 3JH, UK; o_leviuche@hotmail.com; 9Department of Community Medicine, College of Health Sciences, University of Jos, P.M.B. 2084, Jos 930003, Nigeria; minerc@unijos.edu.ng; 10School of Management and Marketing, Curtin Business School, Curtin University, Bentley, WA 6151, Australia; Richard.Oloruntoba@curtin.edu.au; 11Department of Computer Science, University of Jos, P.M.B. 2084, Jos 930003, Nigeria; ishayat@unijos.edu.ng; 12Tanzania Food and Nutrition Center, P.O. Box 977, Dar es Salaam 11101, Tanzania; mischarwe@yahoo.co.uk; 13Department of Community Medicine, Faculty of Medical Sciences, University of Jos, P.M.B. 2084, Jos 930003, Nigeria; envuladue@unijos.edu.ng; 14County Durham and Darlington, National Health Service (NHS) Foundation, Darlington DL3 0PD, UK; o.nwaeze@nhs.net; 15School of Health Science, Western Sydney University, Campbelltown, NSW 2560, Australia

**Keywords:** SARS-CoV-2, sub-Sahara Africa, risks perception

## Abstract

This study investigated risk perception of contracting and dying of SARS-CoV-2 in sub-Sahara Africa during and after the lockdown periods. Two online surveys were conducted one year apart, with participants 18 years and above living in sub-Sahara Africa or the diaspora. Each survey took four weeks. The first survey was taken from 18 April to 16 May 2020, i.e., during the lockdown. The second survey was taken from 14 April to 14 May 2021, i.e., after the lockdown. A cross-sectional study using adopted and modified questionnaires for both surveys were distributed through online platforms. Question about risks perception of contracting and dying of SARS-CoV-2 were asked. The Helsinki declaration was applied, and ethical approvals were obtained. Total responses for both surveys, i.e., both during and after the lockdown, was 4605. The mean age was similar in both surveys (18–28 years). The mean risk perception scores were higher after lockdown by 3.59%. Factors associated with risk perception of COVID-19 were survey period, age group, region of residence, and occupation. Non-health care workers had a lower risk perception of COVID-19. This first comparative study on the level of risk perception of Africans during and after the lockdown shows that one in every three and every four persons in sub-Sahara Africa felt at high risk of contracting COVID-19 and thought they could die from contracting the same, respectively.

## 1. Introduction

Since SARS-CoV-2, a beta coronavirus genre more closely linked to the SARS-CoV-1 (79% sequence identity) than to the MERS-CoV (52% identity) [1], was declared a pandemic by WHO in 2020 [2], the virus has infected over 237 million people, with the death of not less than 4,839,000 people, with US being the worst-affected country recording 727,273 deaths, followed by Brazil—599,414 deaths, India—449,883 deaths, Mexico—279,894 deaths, and Russia—212,625 deaths, as of 7 October 2021. In Africa, South Africa—87,981, Tunisia—24,971, Egypt—17,531, and Morocco—14,390 account for the highest number of deaths from COVID-19 in the region [3]. The SARS-CoV-2 infection presents with dry cough, fever, dyspnea, and lung trouble, among other signs [4]. With no effective cure or current drug for the treatment of the infection in sight, SARS-CoV-2 continues to be a source of concern across the globe and more so in sub-Saharan Africa considering the poor health care system [5]. The rollout of the vaccines has been anything but smooth due to the mixed messages from the various governments and the difficulty in accessibility for developing countries [6,7]. This increases the mistrust displayed by citizens across the globe and increases the perception of risk in the community [7].

In sub-Saharan Africa (SSA), the impact of SARS-CoV-2 has remained minimal compared to the Americas, Europe, and Asia; however, there has been an increase in COVID-19 deaths across Africa since mid-July 2021. Although the reasons for this are not well understood, researchers have suggested that the demographic age structure of sub-Saharan Africa is the leading factor of the low morbidity and mortality of COVID-19 compared to other regions of the world [8]. Other factors, such as the lack of long-term care facilities, potential cross-protection from previous exposure to circulating coronaviruses, and low testing of SARS-CoV-2, have resulted in an undercounting of deaths and effective government public health responses have contributed to the lower burden of the disease [8]. According to data from the US Centers for Disease Control and Prevention (CDC), 80% of COVID-19-related deaths occur in individuals aged 65 years and older [9], with UK data demonstrating that advanced age is the strongest risk for death and dramatically outweighs the risks associated with any other demographic factor or medical condition [10]. The median age of the SSA population is considerably lower than other regions, with a median age of 18 and only 3.0% of the African population older than 65 years [11,12].

South Africa has one of the highest infection and deaths rates due to COVID-19 in Africa. In addition, countries have already implemented the recommended public health regulations, such as strict, partial, or full lockdown procedures [4]; social distancing; mask-wearing in public places; and vaccination rolled out in the majority of the SSA countries. Countries that embarked on total lockdown were avoiding any national resurgence. Wide-scale domestic, foreign, and religious events have been cancelled for fear of SARS-Cov-2 outbreak as they were considered super-spreaders of the virus [13,14]. Such actions have an enormous socio-economic impact on the country [15], and the shutdown has upstretched fears of economic repercussions [16]. Due to this pandemic, everything about human life, including exports and imports of goods, business, infrastructural development, agriculture, and education, seem to have stopped, and these have a direct and indirect negative effect on the economy [17] given the already weak economy of some SSA countries and the resultant drawback risks.

In South Africa, a study showed that a higher perceived risk of COVID-19 infection was associated with greater depressive symptoms and, with such high rates of severe mental illness coupled with the low availability of mental healthcare amidst COVID-19 in the region, there is a need for studies to understand if the change in time has any effect on the level of risk perception for targeted intervention, including the need for immediate and accessible psychological resources [18]. In our recent study conducted during the early lockdown, SSA displayed high individual risk perception scores, which was greater in older participants and those working in health care sectors after adjusting for covariates [19]. It is unclear whether similar risk perceptions and associated factors remain after participants have grown in their knowledge of the disease spread and the commencement of the vaccine rollout in most SSA countries. The current study aims to investigate the individual perception of risk for contracting SARS-Cov-2 and the associated factors by comparing the data obtained during the lockdown with those obtained in the post-lockdown period in SSA. The findings of this study will provide an understanding of the population at higher risk for which can be used to implement emergency policies to counter the spread of SARS-Cov-2.

## 2. Materials and Methods

We conducted two online surveys one year apart in the SSA region, including West Africa, Southern Africa, East Africa, and Central Africa. Participants were aged 18 years and over (*n* = 1005 and *n* = 1004) living in Africa and outside Africa (Diaspora). The first survey was conducted from 18 April–16 May 2020 (during the lockdown), when most of the countries in the region were under mandatory lockdown and restricted movement, and the second survey was conducted between 14 March and14 April 2021(after the lockdown), when most of the mandatory lockdown was over.

### 2.1. Data Collection

An e-link to the survey was disseminated via emails and posted on social media platforms (Facebook and WhatsApp) and was available for four weeks at each period (during the lockdown and after the lockdown). Of the 4605 participants for both the first and second surveys, 4572 provided responses on their place of origin and out of these numbers, 4551 mentioned their countries in SSA and were included in the analysis. Internal procedures during the data collection explained the small difference in the number of participants between the two surveys (*n* = 2001, vs. *n* = 2550). Table 1 provides details about the sociodemographic characteristics of the participants and health status. The questionnaire included a brief overview of the context, purpose, procedures, nature of participation, privacy and confidentiality statements, and notes to be filled out [20].

In order to further reduce bias, this online survey used a Likert scale with provisions for neutral responses so that the answers were not influenced in one way or another. The participants did not receive any incentives; their responses were voluntary and anonymized. Testing for the internal validity of the survey items, the Cronbach’s alpha coefficient score ranged from 0.70 and 0.74, indicating satisfactory consistency.

### 2.2. Measures

The questionnaire collected data on sociodemographic variables (Table 1), self-assessment of risks about COVID-19, and if they thought the public health authorities in their countries were doing enough to contain the virus, whether they or any of their close relatives were affected by COVID-19, and whether or not they thouht COVID-19 is real. Other questions relating to knowledge of COVID-19, habits during the lockdown, and attitudes towards the infection were included in survey 1, while questions related to knowledge and attitude towards COVID vaccination were included in survey 2. Those questions that were not in both surveys are not included in the current analysis, but the interested reader on these topics is referred to the published articles for a description of items and responses.

### 2.3. Assessment of Risks about COVID-19

Self-assessments of risks about COVID-19 were measured with two items which were common in both surveys. The first item concerned the perception of the risk of being infected by COVID-19 (Q1: “Please rate your risk of being infected with the Coronavirus (COVID-19)”), and the second item was the self-assessment of the risk of dying from the infection (Q2: “Please rate your risk of dying from the Coronavirus (COVID-19) infection”). Each question used a Likert scale with five levels [21]. The scores for each item ranged from 0 (lowest) to 4 (highest). The perceived risk towards COVID-19 score ranged from 0–8 points.

### 2.4. Ethical Consideration

These cross-sectional studies were approved by the Human Research Ethics Committee of the Cross River State Ministry of Health, Nigeria (CRSMOH/RP/REC/2020/116) for the first survey, and by the Humanities and Social Sciences Research Ethics Committee (HSSREC 00002504/2021) of the University of KwaZulu-Natal, Durban, South Africa for the second survey. The studies adhered to the principles of the Helsinki declaration (as modified in Fortaleza 2013) for research involving human subjects [22]. Prior to the studies, an explanation detailing the nature and purpose of the studies was provided to all participants using an online preamble. Informed consent was obtained from the participants who were required to answer either a ‘yes’ or ‘no’ to a question on whether or not they were willing to participate in the survey voluntarily. The confidentiality of participants’ responses was assured, and anonymity was maintained. To ensure that only one response per respondent was included in the study per survey, participants were instructed not to take part in the survey at both periods more than once, and, during analysis, we also restricted the data by IP address of the participants.

### 2.5. Statistical Analysis

Continuous variables were summarized using descriptive statistics, including the number of observations used in the calculation (n), mean, and standard deviation (SD), while categorical variables were summarized as counts and percentages of each category for all demographic characteristics for during and post lockdown. To profile the risk of being infected by COVID-19 and the risk of dying from the infection, the Chi-square test was used to determine their prevalence. Each demographic characteristic was compared with a t-test for 2 groups, and one-way analysis of variance (ANOVA) for 3 or more groups. Simple linear regression analysis was used to report the unadjusted coefficient and retained those variables with *p* value <  0.20 in order to build a multiple linear regression analysis. For multiple linear regression, an elimination procedure was applied to remove non-significant variables (*p* > 0.05). All analyses were performed using ‘SVY’ commands in STATA/MP V.13.0 (Stata Corp, College Station, TX, USA).

## 3. Results

### 3.1. Descriptive Statistics

The descriptive statistics of the sociodemographic variables are presented in Table 1, showing the summary of responses from those who participated in the survey during the lockdown and after the lockdown periods. Total responses were a combination of both survey responses. The mean age of the respondents 34.4 ± 11.7 years was similar in both surveys (34.1 ± 11.6 and 34.6 ± 11.8 years, during and post-lockdown respectively). Table 1 shows that most of the respondents were in the 18–28 years age group (38%, *n* = 1697). There was an almost equal representation of male and female respondents. Most respondents (55.7%, *n* = 2522) were not married, about half of them (52.9%, *n* = 2383) had completed post-secondary education, and many were employed (66.9%, *n* = 3001) and worked in a non-healthcare sector (40.5%, *n* = 1602) at the time of the studies. Furthermore, 89.7% of the respondents (*n* = 4042) were Christians.

Figure 1a,b presents the percentage of responses for the items that make up the dependent variable:) the risk of becoming infected with COVID-19 and the risk of dying from COVID-19 infection, respectively. For each item, the proportion from both surveys who felt either at high or very high risk of contracting the infection was 39.9%, and about a quarter thought they were at risk of dying from the infection. Compared with during lockdown, significantly more respondents felt at high risk [17.12%; 95%CI 16.05–18.24% versus 9.27% 95%CI 8.46–10.15] and very high risk [7.21%, 95%CI 6.49–9.00% versus 5.34%, 95%CI 4.72–6.03%] of becoming infected from COVID-19 post-lockdown. Similarly, 11.76% [95%CI 10.85–12.72%] of respondents felt at high risk of dying from COVID-19 infection after the lockdown compared with 4.92% [95%CI 4.33–5.59%] during the lockdown.

### 3.2. Mean Scores and Unadjusted Factors of Risk Perception for Contracting COVID-19

Figure 2 shows the mean scores for the perceived risk of COVID-19 at 95% CIs (presented as error bars). SSA respondents had significantly higher mean risk perception scores after the COVID-19 lockdown compared with during the lockdown period (*p* < 0.0005). The perceived risk estimated from the second survey was 0.49 higher than that of the lockdown period. This translates to a Cohen’s D value of 0.21 SD (i.e., the mean of survey 2 and survey 1, and the pooled standard deviation for the entire sample) [23] which was higher than the mean scores of the perceived risk of COVID-19 during the lockdown. From the Emslie data [24], respondents who participated in the post lockdown survey were 58% more likely to perceive a risk of contracting or dying from COVID-19 compared with those that participated in the survey during the lockdown period. This is clinically significant [24].

Table 2 presents the mean risk perception scores as well as the unadjusted odds ratios and their 95% CIs for factors associated with risk perception by the demographic characteristics. Data presented were pooled from both surveys. The mean risk perception scores were significantly different between the study periods. Compared with the lockdown period, the results indicated that perceived risk scores for contracting COVID-19 post-lockdown increased by 0.49 (95%CI 0.36, 0.63, *p* < 0.0001) and increased with age. Respondents aged above 28 years had significantly higher risk perceptions scores compared with those aged 18–28 years.

Other factors associated with perceived risk scores for contracting COVID-19 in the unadjusted analysis are region of origin, marital, educational and working status, and respondents’ occupation.

### 3.3. Factors Associated with Perceived Risk for Contracting COVID-19 during Lockdown and Post-Lockdown

Table 3 shows the adjusted coefficients (β) with 95% CIs of the factors influencing perceived risk for contracting COVID-19 during and post-lockdown period in SSA countries. After adjusting for potential confounding factors, the post-lockdown period and age > 28 years were significantly associated with increased risk perception. Respondents from East and Southern Africa reported higher risk perception scores compared with those from West Africa. Working in a non-healthcare sector (β −0.56, 95% CI −0.73, −0.38) and being a student (β −0.60, 95% CI −0.82, −0.38) were associated with a reduction in the risk perception scores for contracting COVID-19.

## 4. Discussion

To the best of the authors’ knowledge, this is the first study to compare the level of risk perception of Africans during and after the COVID-19 lockdown period. The study found that more than one in every three persons in this SSA sample and about one in every four respondents felt at high risk of contracting COVID-19 and thought they could die if they contracted COVID-19, respectively. Compared with a pre-lockdown period, respondents who participated in the post-lockdown survey reported a significantly higher risk of COVID-19, particularly the older people and respondents that lived in East and Southern Africa.. The perceived risk of contracting COVID-19 increased significantly between the two surveys showing that respondents overestimated their chances of contracting or dying from COVID-19 by 58%. Although such finding does not reflect a strong deviation from rational behavior, it is common in the literature [25,26], and the likelihood of overestimating small risks fatalities occurs rationally in a Bayesian model when learning is based on partial information [27]. Furthermore, those who worked in health care sectors reported higher risk perception of COVID-19 whereas students who participated in the survey after the lockdown reported lower risk perception compared with other groups.

This increase in risk perception after the lockdown which was found in this study may be attributed to various factors, including the rise in the COVID-19 infections and related deaths in the region after the lockdown [26]. In addition, the controversies surrounding the rolling out of the COVID-19 vaccine globally and the uncertainty of vaccine acceptance in the region [28]. Despite various governments’ efforts at increased sensitization of the populace on the disease, their inability to answer the questions raised about the COVID-19 vaccine [29] could have necessitated the increased risk perception of getting infected after the lockdown.

Past repeat studies have found differences in risk perception of COVID-19 over a time period. A fourth-round survey of respondents in Kenya, East Africa demonstrated that the perceived risk of coronavirus remained about the same, but the proportion that said they were at high risk because they interacted with a lot of people every day more than doubled (from 20% to 54%) [30]. In France, two successive representative surveys, one conducted about 2 weeks after the lockdown started, and the other about 2 weeks before the lockdown ended, found significantly higher risk perception in the second survey than in the first survey. The authors attributed the comparative pessimism in survey 2 to a concomitant increase in the respondents’ perceived chances to contract the disease and a decreased expected prevalence rate [26].

In the present study, we found that older age (≥28 years) was associated with an increased risk of susceptibility to COVID-19; this was consistent with past studies [29,31,32,33] which showed that older individuals had a higher risk perception of contracting the infection and were more likely to develop more severe complications of COVID-19 or die compared with the younger individuals [32]. The sigh of relief brought about by the post-lockdown era had a serious effect on the younger age groups who, at that time, had a lower risk perception for contracting the infection, as seen in a study by Dillard et al. [30]. The perceived low risk of infection by younger respondents may make them less cooperative and less compliant with the safety measures [34], thus encouraging the spread of the virus while putting a greater part of the population at risk of COVID-19 infections [29,30]. This finding could be attributable to the fact that younger people are the more active age group in any given population.

In his write up about medical students during this COVID-19 era, Flaxman et al. stated that students are not essential workers [35], which implies that they are not yet classified as healthcare workers since they are not paid or tasked with the responsibility of patient care in healthcare facilities [36]. In this study, people working in non-healthcare sectors, including students, felt less susceptible to the infection. This finding can be attributed to exposure of healthcare workers to infected people; the absence of personal protective equipment, particularly in SSA countries [37]; over crowdedness of medical facilities; and inadequate provision of needed health management instruments [37]. There is a need for regular educational intervention and training programs on infection control practices for COVID-19 across all healthcare professions.

The study also found regional differences in the level of perceived risk for contracting COVID-19 during and after the lockdown. Although the risk perception scores were reduced after the lockdown among East Africans, they and Southern Africans felt at greater risk of COVID-19 infection compared with West African respondents. Such regional differences with regard to COVID-19 infection was reported to vary from location to location with significantly varying degrees of impact [38]. In this study, we noted that respondents from two of the participating SSA countries reported higher risk perception scores for contracting the virus, and, for the other two regions—Western and Central Africa—the risk perception remained unchanged. Although a cross-sectional study from China did not find a significant regional variation in the risk perception of the SARS-CoV-2 pandemic [39], there are factors that come to play, including the cultural beliefs and inclinations of the people, their religious orientations, the governmental policies in place for the control of the spread of the disease, and the individual tendencies for survival among many others [40].

The study has some limitations which should be interpreted within the context of the study. Using a perceived risk score that ranged from 0–8 points may violate some linear regression assumptions [19,41]. The use of an online survey has the potential to result in selection bias and could have unduly excluded residents in SSA without internet access. The preponderance of educated persons in this study is another limitation that is a characteristic of most survey studies in Africa [19,20,37,42] and elsewhere [43]. These study findings may not be generalizable to the entire SSA because not all countries in SSA answered the questions. Besides these limitations, this is the first study to compare the level of risk perception of Africans during and after the COVID-19 lockdown period.

## 5. Conclusions

It is clear that, during the lockdown, people had some measure of certainty regarding the SARS-CoV-2, which dissipated after the lockdown as the rates of infection across the globe, particularly in SSA, were seen to be on the surge with a reported increasing number of deaths. Notably, the factors influencing risk perception scores remained the same during and after the lockdown and this included age, region of origin, and occupation. The rollout of the COVID-19 vaccine and the controversies regarding the effectiveness of the vaccines, as well as the media focus on the new variant, may have heightened the perceived risk of infection. There is the need for governments in SSA to intensify the public awareness of the emergence of new variants of the virus and design compatible ways of ensuring that the vaccines are at the reach of everyone and that everyone should be encouraged to receive his/her shot of the vaccine to stay safe and alive. Furthermore, further studies need to be carried out to ascertain the post lockdown risk perception, since from existent studies, there seems to be non-availability of data on the post lockdown risk perception of contracting SARS-CoV-2 and with the ongoing vaccination in view.

## Figures and Tables

**Figure 1 ijerph-18-11091-f001:**
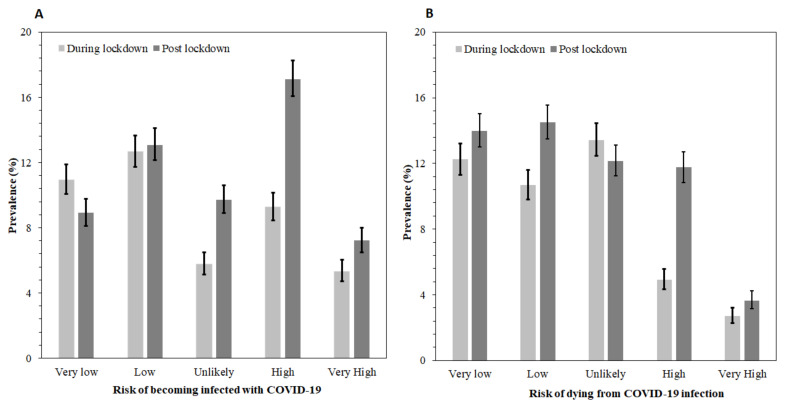
Proportion of responses for perceived risks of COVID-19: (**A**), the risk of becoming infected with COVID-19; (**B**) the risk of dying from COVID-19 infection Error bars are 95% confidence intervals. Unlikely means no risk.

**Figure 2 ijerph-18-11091-f002:**
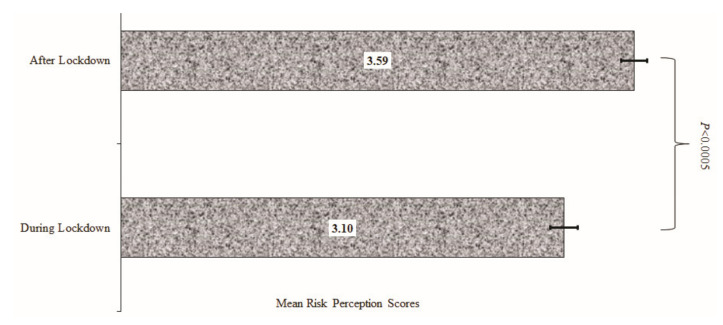
Mean score for the perceived risk of contracting COVID-19 after and during lockdown.

**Table 1 ijerph-18-11091-t001:** Sociodemographic characteristics of respondents in both surveys.

Demographics	Total (N = 4551)	During Lockdown (*n* = 2001)	Post-Lockdown (*n* = 2550)
*Age category in years*			
18–28 years	1697 (38.0)	774(39.1)	923 (37.2)
29–38	1242 (27.8)	526 (26.5)	716 (28.9)
39–48	939 (21.1)	439 (22.2)	500 (20.2)
49+ years	584 (13.1)	242 (12.2)	342 (13.8)
*Sex*			
Males	2467 (54.5)	1095 (55.2)	1372 (53.8)
Females	2057 (45.5)	889 (44.8)	1168 (45.8)
*SSA Region of Origin*			
West Africa	2572(56.5)	1122 (56.1)	1450 (56.9)
East Africa	347(7.6)	212 (10.6)	135 (5.3)
Central Africa	570 (12.5)	253 (12.6)	317 (12.4)
Southern Africa	1062 (23.3)	414 (20.7)	648 (25.4)
*Country of residence*			
Africa	4250 (93.6)	1852 (92.6)	2398 (94.4)
Diaspora	291 (6.4)	149 (7.4)	142 (5.6)
*Marital Status*			
Married/de facto	2003 (44.3)	876 (44.1)	1127 (44.4)
Not married ^†^	2522 (55.7)	1112 (55.9)	1410 (55.6)
*Educational status*			
Master’s degree or more ^‡^	1383 (30.7)	639 (32.1)	744 (29.5)
Bachelor’s degree ^α^	2383 (52.9)	1086 (54.6)	1297 (51.5)
Secondary/primary	741 (16.4)	264 (13.3)	477 (19.0)
*Working status*			
Employed/self employed	3001 (66.9)	1353 (68.0)	1648 (65.9)
Unemployed/retired	1488 (33.1)	636 (32.0)	852 (33.1)
*Religion*			
Christianity	4042 (89.7)	1758 (88.4)	2284 (90.8)
Others ^ᵖ^	462 (10.3)	230 (11.6)	232(9.2)
*Occupation ^β^*			
Healthcare sector	1240 (31.5)	443 (24.3)	797 (37.6)
Non-healthcare	1602 (40.6)	1014 (55.7)	588 (27.7)
Student	1099 (27.9)	364 (20.0)	735 (34.7)

†, divorced, separated, widowed and single; ‡ included Masters and PhD; postgraduate, α, diploma and bachelor degree; ᵖ, included Muslims and African traditionalist; *β = no response from 610* respondents for this variable (13.4%). SD = standard deviation. Values are numbers (%) except for mean age.

**Table 2 ijerph-18-11091-t002:** Mean scores and unadjusted coefficients (B) for factors associated with perceived risk of contracting COVID-19 during lockdown and post-lockdown.

Variables	Mean Scores (±SD)	B [95%CI]	*p*-Value
** *Survey period* **			
Period 1 (during lockdown)	3.10 (2.19)	Ref	
Period 2 (post-lockdown)	3.59 (2.36)	0.49 [0.36, 0.62]	<0.001
Demography			
*Age category in years*			
18–28 years	3.13 (2.24)	Ref	
29–38	3.51 (2.31)	0.38 [0.22, 0.55]	<0.001
39–48	3.57 (2.35)	0.44 [0.26, 0.63]	<0.001
49+ years	3.58 (2.30)	0.45 [0.23, 0.66]	<0.001
*Sex*			
Males	3.42 (2.32)	Ref	
Females	3.34 (2.27)	−0.08 [−0.22, 0.005]	0.226
*SSA Region of Origin*			
West Africa	3.26 (2.24)	Ref	
East Africa	3.78 (2.41)	0.51 [0.26, 0.77]	<0.001
Central Africa	3.22 (2.44)	−0.05 [−0.26, 0.16]	0.658
Southern Africa	3.61 (2.30)	0.35 [0.19, 0.52]	<0.001
*Country of residence*			
Africa	3.39 (2.30)	Ref	
Diaspora	3.26(2.23)	−0.13 [−0.40, 0.15]	0.360
*Marital status*			
Married	3.52(2.30)	Ref	
Not married	3.27(2.30)	−0.25 [−0.39, −0.12]	<0.001
*Educational status*			
Master’s degree or more	3.50(2.25)	Ref	
Bachelor’s degree	3.37(2.32)	−0.13 [−0.28, 0.02]	0.089
Secondary/Primary	3.20 (2.32)	−0.31 [−0.51, −0.10]	0.004
*Working status*			
Employed/self employed	3.54 (2.30)	Ref	
Unemployed/retired	3.10 (2.26)	−0.43 [−0.57, −0.29]	<0.001
*Religion*			
Christianity	3.37(2.30)	Ref	0.676
Others	3.42(2.29)	0.05 [−0.17, 0.27]	
*Occupation*			
Healthcare sector	3.83 (2.34)	Ref	
Non-healthcare	3.20 (2.23)	−0.63 [−0.80, −0.46]	<0.001
Student	3.09 (2.24)	−0.75 [−0.93, −0.56]	<0.001

SD, standard deviation; CI, confidence interval that do not include 0.00 were significant. SSA, sub-Saharan Africa; Ref, reference (0.00).

**Table 3 ijerph-18-11091-t003:** Factors associated with perceived risk of contracting COVID-19 in sub-Saharan Africa.

Variables	β [95%CI]	*p*-Value
** *Year of survey* **		
Period 1 (during lockdown)	Ref	
Period 2 (post-lockdown)	0.42 [0.27, 0.57]	<0.001
Demography		
*Age category in years*		
18–28 years	Ref	
29–38	0.25 [0.04, 0.46]	0.020
39–48	0.31 [0.08, 0.54]	0.010
49+ years	0.31 [0.05, 0.58]	0.020
*SSA Region of Origin*		
West Africa	Ref	
East Africa	0.55 [0.28, 0.82]	<0.001
Central Africa	0.08 [−0.15, 0.31]	0.490
Southern Africa	0.37 [0.19, 0.54]	<0.001
*Occupation*		
Healthcare sector	Ref	
Non-healthcare	−0.56 [−0.73, −0.38]	<0.001
Student	−0.60 [−0.82, −0.38]	<0.001

CI, confidence interval that does not include 0.00 were significant. SSA, sub-Saharan Africa; Ref, reference (0.00). β is the adjusted coefficient from the linear regression model.

## Data Availability

The datasets analyzed during this study are available from the authors on reasonable request.

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
