# Peer review of "Differences in Perceived Risk of Contracting SARS-CoV-2 during and after the Lockdown in Sub-Saharan African Countries"

_ijerph, 2021, doi:10.3390/ijerph182111091_

Round 1

Reviewer 1 Report

1. The authors made an inference about the SSA population from a sample survey that 1/3 felt at high risk of contracting COVID-19 and 1/4 thought they could die if contracting COVID-19. However, as acknowledged by the authors in the Discussion section, because the demographics of participants of the sample survey differ from that of the SSA population (i.e., not representative), these estimates may not be generalized to the entire SSA. To provide readers with a more informed judgement about how well these estimates could reflect or represent the entire SSA population, my suggestion for the authors is to provide (or cite) the distributions of demographics of the SSA population (e.g., age, countries/areas, education, etc); such distributions should be available through census related surveys that countries or regions regularly conduct. This will help readers to see how the demographics of the sample survey is similar or different from the SSA population when they make interpretation of results of this study.

2. A major conclusion of the authors is that the perceived risk estimated from the second survey (1 year apart from the lockdown) is higher than that during the lockdown, shown by the mean risk perception scores (i.e., 3.59 after the lockdown vs. 3.10 during the lockdown). The difference is statistically significant; however, my concern/question is: Is such a difference (3.59 vs. 3.10) clinically or psychologically meaningful and how? What does the 0.49 difference between the two scores really mean, clinically or psychologically? I suggest the authors adding a discussion point about the meaning of this difference, preferably with relevant citations. It is not uncommon to gain statistical significance over a small difference when the sample size is big enough; what is equally, if not more, important is how such a difference is clinically or psychologically relevant. 

3. Please provide citations for the Likert scale in the Method section. 

4. In the Discussion section, lines 286-288: the comparison to the U.S. study is questionable: to my understanding, the U.S. study (cited by the authors [34]) reported actual risk of infection (not perceived risk), which is not directly comparable to the perceived risk under the current study. Even for the same population, actual risk could differ a lot from perceived risk because for various reasons. Such a comparison cannot support the authors' argument. In case I misunderstand anything, please clarify. 

5. In the Discussion section, lines 266-269, the authors mentioned that "two successive representative surveys conducted during the lockdown also showed significantly higher risk perception in the second survey" in a French study (cited by the authors [25]). While the risk perception in the second survey is higher than the first, both of the French surveys were conducted during the lockdown, not 1 during 1 after the lockdown, which is quite different from the current study where the authors had 1 year part after the lockdown. The French study is not comparable enough to the current study for the authors to cite as a study with similar aims and findings. The authors may cite the French study in a more appropriate way. 

6. In Figure 1, the horizontal axis: why is "unlikely" in the middle of the scale? Does "unlikely" mean a risk of zero, or medium risk? If "unlikely" means zero risk, wouldn't it be better to place it left to the "very low" risk? If "unlikely" means medium risk, then please clarify and maybe name it in another way to avoid confusion. 

Author Response

Reviewer 1

  1. The authors made an inference about the SSA population from a sample survey that 1/3 felt at high risk of contracting COVID-19 and 1/4 thought they could die if contracting COVID-19. However, as acknowledged by the authors in the Discussion section, because the demographics of participants of the sample survey differ from that of the SSA population (i.e., not representative), these estimates may not be generalized to the entire SSA. To provide readers with a more informed judgement about how well these estimates could reflect or represent the entire SSA population, my suggestion for the authors is to provide (or cite) the distributions of demographics of the SSA population (e.g., age, countries/areas, education, etc); such distributions should be available through census related surveys that countries or regions regularly conduct. This will help readers to see how the demographics of the sample survey is similar or different from the SSA population when they make interpretation of results of this study.

Reply: Thanks for this comment. We apologise for the error in our interpreted result. This has been revised as it was meant to refer to the survey sample and not the SSA population.

‘The study found that more than one in every three persons in this SSA sample and about one in every four respondents felt at high risk of contracting COVID-19’

  1. A major conclusion of the authors is that the perceived risk estimated from the second survey (1 year apart from the lockdown) is higher than that during the lockdown, shown by the mean risk perception scores (i.e., 3.59 after the lockdown vs. 3.10 during the lockdown). The difference is statistically significant; however, my concern/question is: Is such a difference (3.59 vs. 3.10) clinically or psychologically meaningful and how? What does the 0.49 difference between the two scores really mean, clinically or psychologically? I suggest the authors adding a discussion point about the meaning of this difference, preferably with relevant citations. It is not uncommon to gain statistical significance over a small difference when the sample size is big enough; what is equally, if not more, important is how such a difference is clinically or psychologically relevant. 

Reply: Thanks for this suggestion. We have now included the clinical implication of the mean difference in perceived risk found in this study by calculating the Cohen d in relation to previous findings. This was included in the results and re-iterated in the discussion

 = 0..21.

The section in the results now reads:

“The perceived risk estimated from the second survey was 0.49 higher than that of the lockdown period which translates to a Cohen d value of 0.21 SD (i.e. mean of survey 2-mean of survey 1/pooled standard deviation for the entire sample)[3] higher than the mean scores of perceived risk of COVID-19 during the lockdown. From the Emslie data[4], respondents who participated in the post lockdown survey were 58% more likely to perceive a risk of contracting or dying from COVID-19 compared with those that participated in the survey during the lockdown period and this is clinically significant[4].”

And in the discussion

“The perceived risk of contracting COVID-19 increased significantly between the two surveys showing that respondents overestimated their chances of contracting or dying from COVID-19 by 58%. Although such finding does not reflect a strong deviation from rational behavior, it is common in the literature[5, 6] and the likelihood of overestimating small risks fatalities occurs rationally in a Bayesian model when learning is based on partial information[7]. Furthermore, those who worked in health care sectors reported higher risk perception of COVID-19 whereas students who participated in the survey after the lockdown reported lower risk perception compared with other groups”

 ‘3. Please provide citations for the Likert scale in the Method section. 

Reply: done

  1. In the Discussion section, lines 286-288: the comparison to the U.S. study is questionable: to my understanding, the U.S. study (cited by the authors [34]) reported actual risk of infection (not perceived risk), which is not directly comparable to the perceived risk under the current study. Even for the same population, actual risk could differ a lot from perceived risk because for various reasons. Such a comparison cannot support the authors' argument. In case I misunderstand anything, please clarify. 

Reply: Thanks for this. We agree with the reviewer and have now removed the reference.

  1. In the Discussion section, lines 266-269, the authors mentioned that "two successive representative surveys conducted during the lockdown also showed significantly higher risk perception in the second survey" in a French study (cited by the authors [25]). While the risk perception in the second survey is higher than the first, both of the French surveys were conducted during the lockdown, not 1 during 1 after the lockdown, which is quite different from the current study where the authors had 1 year part after the lockdown. The French study is not comparable enough to the current study for the authors to cite as a study with similar aims and findings. The authors may cite the French study in a more appropriate way. 

Reply: Thanks for the comment. The section has been revised to read: Past repeat studies have found differences in risk perception of COVID-19 over a time period. A fourth-round survey…. In France, two successive representative surveys, one conducted about 2 weeks after lockdown started, and the other about 2 weeks before lockdown ended, found significantly higher risk perception in the second survey than in the first survey. The authors attributed the comparative pessimism in survey 2 to a concomitant increase in the respondents’ perceived chances to contract the disease and a decreased expected prevalence rate.

  1. In Figure 1, the horizontal axis: why is "unlikely" in the middle of the scale? Does "unlikely" mean a risk of zero, or medium risk? If "unlikely" means zero risk, wouldn't it be better to place it left to the "very low" risk? If "unlikely" means medium risk, then please clarify and maybe name it in another way to avoid confusion. 

Reply: Figure 1 has been revised as suggested. Unlikely mean a risk of zero and this was added in the legend [Figure 1. Proportion of responses for perceived risks of COVID-19. Error bars are 95% confidence intervals. Unlikely means no risk]

Reviewer 2 Report

In general it is a very interesting paper. I only have a few comments.

- Table 1, is a bit confusing. Put the same number of decimal places.

- Increase the size of figure 1. On the y-axis, what are%? Specify it.

- Update the bibliography according to the regulations of the journal

Author Response

Reviewer 2

Comments and Suggestions for Authors

In general it is a very interesting paper. I only have a few comments.

- Table 1, is a bit confusing. Put the same number of decimal places.

Reply: Thanks. We have revised the table by removing the mean age to minimise confusion with the age in categories. The mean age was placed in the text and the first column was formatted to include the number of respondents.

- Increase the size of figure 1. On the y-axis, what are%? Specify it.

Reply: % has been included in the y axis and legend revised. The size of the figure has been increased and high resolution version uploaded

- Update the bibliography according to the regulations of the journal

Reply: Done
